# Towards Multilingual Interlinear Morphological Glossing

**Shu Okabe**
Université Paris-Saclay & CNRS
LISN, rue du Belvédère
91405 Orsay, France
shu.okabe@limsi.fr

**François Yvon**
Sorbonne Université & CNRS
ISIR, 5 Place Jussieu
75005 Paris, France
francois.yvon@isir.upmc.fr

## Abstract

Interlinear Morphological Glosses are annotations produced in the context of language documentation. Their goal is to identify morphs occurring in an L1 sentence and to explicit their function and meaning, with the further support of an associated translation in L2. We study here the task of automatic glossing, aiming to provide linguists with adequate tools to facilitate this process. Our formalisation of glossing uses a latent variable Conditional Random Field (CRF), which labels the L1 morphs while simultaneously aligning them to L2 words. In experiments with several under-resourced languages, we show that this approach is both effective and data-efficient and mitigates the problem of annotating unknown morphs. We also discuss various design choices regarding the alignment process and the selection of features. We finally demonstrate that it can benefit from multilingual (pre-)training, achieving results which outperform very strong baselines.

## 1 Introduction

Interlinear Morphological Gloss (IMG) (Lehmann, 2004; Bickel et al., 2008) is an annotation layer aimed to explicit the meaning and function of each morpheme in some documentation ('object') language L1, using a (meta-)language L2. In computational language documentation scenarios, L1 is typically a low-resource language under study, and L2 is a well-resourced language such as English.

Figure 1 displays an example IMG: the source sentence $t$ in L1 is overtly segmented into a sequence of morphemes ($x$), each of which is in one-to-one correspondence the corresponding gloss sequence $y$. Each unit in the gloss tier is either a grammatical description (OBL for the *oblique* marker in Figure 1) or a semantic tag (son in Figure 1), expressed by a lexeme in L2. An idiomatic free translation $z$ in L2 is usually also provided. $y$ and $z$ help linguists unfamiliar with L1 to understand the morphological analysis in $x$.

| $t$ | Nesis | ɬˤono | uži | zown |
|-----|-------|-------|-----|------|
| $x$ | nesi–s | ɬˤono | uži | zow–n |
| $y$ | he.OBL–GEN1 | three | son | be.NPRS–PST.UNW |
| $z$ | He had three sons. | | | |

Figure 1: A sample entry in Tsez: L1 sentence ($t$), and its morpheme-segmented version ($x$), its gloss ($y$), and a L2 translation ($z$). Grammatical glosses are in small capital, lexical glosses in straight orthography.

In this paper, we study the task of automatically computing the gloss tier, assuming that the morphological analysis $x$ and the free L2 translation $z$ are available. As each morpheme has exactly one associated gloss,[1] an obvious formalisation of the task that we mostly adopt views glossing as a *sequence labelling task* performed at the morpheme level. Yet, while grammatical glosses effectively constitute a finite set of labels, the diversity of lexical glosses is unbounded, meaning that our tagging model must accommodate an open vocabulary of labels. This issue proves to be the main challenge of this task, especially in small training data regimes.

To handle such cases, we assume that lexical glosses can be directly inferred from the translation tier, an assumption we share with (McMillan-Major, 2020; Zhao et al., 2020). In our model, we thus consider that the set of possible morpheme labels in any given sentence is the union of (i) all grammatical glosses, (ii) lemmas occurring in the target translation, and (iii) frequently-associated labels from the training data. This makes our model a hybrid between sequence tagging (because of (i) and (iii)) and unsupervised sequence alignment (because of (ii)), as illustrated in Figure 2. Our implementation relies on a variant of Conditional Random Fields (CRFs) (Lafferty et al., 2001; Sut-

---

[1] The reality is slightly more complex, as illustrated by 'compound' glosses such as 'he.OBL' in Figure 1, associating two descriptions to the same morpheme. In this work, such compound labels are processed just as any other purely lexical gloss during feature extraction, training and inference. This allows us to only distinguish two types of labels.

ton and McCallum, 2007), which handles latent variables and offers the ability to locally restrict the set of possible labels. The choice of a CRF-based approach is motivated by its notable data-efficiency, while methods based on neural networks have difficulties handling very low resource settings—this is again confirmed by results in §5.1.

In this work, we generalise previous attempts to tackle this task with sequence tagging systems based on CRFs such as (Moeller and Hulden, 2018; McMillan-Major, 2020; Barriga Martínez et al., 2021) and makes the following contributions: (a) we introduce (§2) a principled and effective end-to-end solution to the open vocabulary problem; (b) we design, implement and evaluate several variants of this solution (§3), which obtain results that match that of the best-performing systems in the 2023 Shared Task on automatic glossing (§5.1); (c) in experiments with several low-resource languages (§4), we evaluate the benefits of an additional multilingual pre-training step, leveraging features that are useful cross-linguistically (§5.5). Owing to the transparency of CRF features, we also provide an analysis of the most useful features (§5.6) and discuss prospects for improving these techniques.

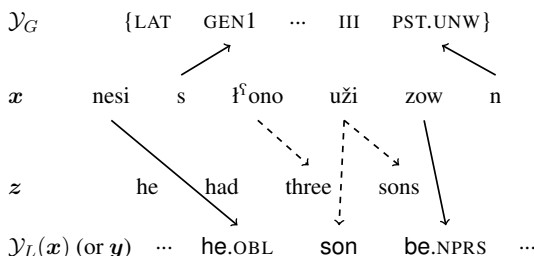

Figure 2: Tagging morphs for the L1 sentence in Fig. 1. $\mathcal{Y}_G$ represents the set of all grammatical glosses in the training data, $\boldsymbol{z}$ the words occurring in the translation, $\mathcal{Y}_L(\boldsymbol{x})$ the set of lexical labels from the training dictionary, and $\boldsymbol{y}$ the reference lexical labels seen in training. During training, automatic alignments between $\boldsymbol{x}$ and $\boldsymbol{z}$ are used. Dashed lines symbolise the ambiguous origin of the label, possibly in both $\boldsymbol{z}$ and $\mathcal{Y}_L(\boldsymbol{x})$.

## 2 A hybrid tagging / alignment model

### 2.1 The tagging component

The core of our approach is a CRF model, the main properties of which are defined below. Assuming for now that the set of possible glosses is a closed set $\mathcal{Y}$, our approach defines the conditional probability of a sequence $\boldsymbol{y}$ of $T$ labels in $\mathcal{Y}$ given a sequence $\boldsymbol{x}$ of $T$ morphemes as:

$$p_{\boldsymbol{\theta}}(\boldsymbol{y}|\boldsymbol{x}) = \frac{1}{Z_{\boldsymbol{\theta}}(\boldsymbol{x})} \exp\left\{\sum_{k=1}^{K} \theta_k G_k(\boldsymbol{x}, \boldsymbol{y})\right\}, \quad (1)$$

where $\{G_k, k = 1 \ldots K\}$ are the feature functions with associated weights $\boldsymbol{\theta} = [\theta_1 \ldots \theta_K]^T \in \mathbb{R}^K$, and $Z_{\boldsymbol{\theta}}(\boldsymbol{x})$ is the partition function summing over all label sequences. For tractability, in linear-chain CRFs, the feature function $G_k$ only test *local* properties, meaning that each $G_k$ decomposes as $G_k(\boldsymbol{x}, \boldsymbol{y}) = \sum_t g_k(y_t, y_{t-1}, t, \boldsymbol{x})$ with $g_k()$ a local feature. Training is performed by maximising the regularised conditional log-likelihood on a set of fully labelled instances, where the regulariser is proportional to the $\ell_1$ ($|\theta|$) or $\ell_2$ ($||\theta||^2$) norm of the parameter vector. Exact decoding of the most likely label sequence is achieved with Dynamic Programming (DP); furthermore, an adaptation of the forward-backward algorithm computes the posterior distribution of any $y_t$ conditioned on $\boldsymbol{x}$.

Using CRFs for sequence labelling tasks has long been the best option in the pre-neural era, owing to (i) fast and data-efficient training procedures, even for medium-size label sets (e.g. hundreds of labels (Schmidt et al., 2013)) and higher-order label dependencies (Vieira et al., 2016), (ii) the ability to handle extremely large sets of interdependent features (Lavergne et al., 2010). They can also be used in combination with dense features computed by deep neural networks (Lample et al., 2016).

### 2.2 Augmenting labels with translations

One of the challenges of automatic glossing is the need to introduce new lexical glosses in the course of the annotation process. This requires extending the basic CRF approach and incorporating a growing repertoire of lexical labels. We make the assumption [H] *that these new labels can be extracted from the L2 translation ($\boldsymbol{z}$ in Figure 1).*

Informally, this means that the grammatical label set $\mathcal{Y}_G$ now needs to be augmented with L2 words in $\boldsymbol{z}$ or equivalently, with indices in $[1 \ldots |\boldsymbol{z}|]$. This raises two related questions: (a) how to exactly specify the set of labels $\mathcal{Y}$ in inference and training. (b) depending on answers to question (a), how to learn the model parameters?

In our model, we additionally consider an extra source of possible lexical labels, $\mathcal{Y}_L(\boldsymbol{x})$, which contains likely glosses for morphemes in $\boldsymbol{x}$. There are several options to design $\mathcal{Y}_L(\boldsymbol{x})$: for instance, to include all the lexical glosses seen in training or to restrict to one or several glosses for each word

$x_t$. In our experiments (§5), we select for each morpheme in $\boldsymbol{x}$ the most frequently associated gloss in the training corpus. $\mathcal{Y}$ thus decomposes into a global part $\mathcal{Y}_G$ and a sentence-dependent part $\mathcal{Y}_L(\boldsymbol{x}) \cup [1 \ldots |\boldsymbol{z}|]$. Performing inference with this model yields values $y_t$ that either directly correspond to the desired gloss or correspond to an integer, in which case the (lexical) gloss at position $t$ is $z_{y_t}$. Formally, the gloss labels are thus obtained as $\tilde{y}_t = \phi(y_t)$, with $\phi()$ the deterministic decoding function defined as $\forall y \in \mathcal{Y}_G \cup \mathcal{Y}_L(\boldsymbol{x}) : \phi(y) = y$ and $\forall i \in [1 \ldots |\boldsymbol{z}|] : \phi(i) = z_i$.

Training this hybrid model is more difficult than for regular CRFs, for lack of directly observing $y_t$. We observe instead $\tilde{y}_t = \phi(y_t)$: while the correspondence is non-ambiguous for grammatical glosses, there is an ambiguity when $\tilde{y}_t$ is present in both $\mathcal{Y}_L(\boldsymbol{x})$ and $\boldsymbol{z}$, or when it occurs multiple times in $\boldsymbol{z}$. We thus introduce a new, partially observed, variable $o_t$ which indicates the origin of gloss $\tilde{y}_t$: $o_t = 0$ when $\tilde{y}_t \in \mathcal{Y}_G \cup \mathcal{Y}_L(\boldsymbol{x})$ and $o_t > 0$ when $\tilde{y}_t$ comes from L2 word $z_{o_t}$. The full model is:

$$p_{\boldsymbol{\theta}}(\boldsymbol{y}, \boldsymbol{o}|\boldsymbol{x}, \boldsymbol{z}) = \frac{1}{Z_{\boldsymbol{\theta}}(\boldsymbol{x}, \boldsymbol{z})} \exp \left\{ \boldsymbol{\theta}^T \boldsymbol{G}(\boldsymbol{x}, \boldsymbol{y}, \boldsymbol{z}, \boldsymbol{o}) \right\}.$$

By making the origin of the lexical label(s) explicit, we distinguish in §3.3 between feature functions associated with *word occurrences* in $\boldsymbol{z}$ and those for *word types* in $\mathcal{Y}_L(\boldsymbol{x})$ (Täckström et al., 2013).

Learning $\theta$ with (partially observed) variables is possible in CRFs and yields a non-convex optimisation problem (see e.g. (Blunsom et al., 2008)). In this case, the gradient of the objective is a difference of two expectations (Dyer et al., 2011, eq. (1)) and can be computed with forward-backward recursions. We, however, pursued another approach, which relies on an automatic word alignment $\boldsymbol{a}$ between lexical glosses and translation (§3.1) to provide proxy information for $\boldsymbol{o}$. Assuming $a_t = 0$ for grammatical glosses and unaligned lexical glosses and $a_t > 0$ otherwise, we can readily derive the supervision information $o_t$ needed in training, according to the heuristics detailed in Table 1, which depend on the values of $y_t$ and $a_t$.

Three heuristic supervision schemes are in Table 1, which vary on how ambiguous label sources are handled: (s1) only considers dictionary entries, which makes the processing of unknown words impossible; (s2) only considers translations, possibly disregarding correct supervision from the dictio-

[2]I.e., the correct label is always part of the search space.

| $y_t$ | $a_t$ | $o_t$ | Sys |
|---|---|---|---|
| $\in \mathcal{Y}_G$ | $= 0$ | 0 | all |
| $\in \mathcal{Y}_L(\boldsymbol{x})$ | $= 0$ | 0 | all |
| | $> 0$ | 0 | (s1) |
| | $> 0$ | $a_t$ | (s2) |
| | $> 0$ | $a_t$ if $y_t = z_{a_t}$ 0 otherwise | (s3) |
| $\notin \mathcal{Y}_L(\boldsymbol{x})$ | $= 0$ | 0 (*) | all |
| | $> 0$ | $a_t$ | all |

Table 1: Three supervisions for the hybrid CRF model. (*) means that the correct label does not occur in the dictionary nor in the translation. To preserve reference reachability,[2] we augment $\mathcal{Y}_L(\boldsymbol{x})$ with the correct label.

nary; (s3) assumes that the label originates from the translation only if an exact match is found.

## 3 Implementation choices

### 3.1 Aligning lexical glosses with target words

To align the lexical glosses with the L2 translation, we use SimAlign (Jalili Sabet et al., 2020), an unsupervised, multilingual word aligner which primarily computes source / target alignment links based on the similarity of the corresponding embeddings in some multilingual space. Note that our task is much simpler than word alignment in bitexts, as the lexical gloss and the translation are often in the same language (e.g. English or Spanish), meaning that similarities can be computed in a monolingual embedding space. We extract alignments from the similarity matrix with `Match` heuristic, as it gave the best results in preliminary experiments. `Match` views alignment as a maximal matching problem in the weighted bipartite graph containing all possible links between lexical glosses and L2 words. This ensures that all lexical morphemes are aligned with exactly one L2 word.[3]

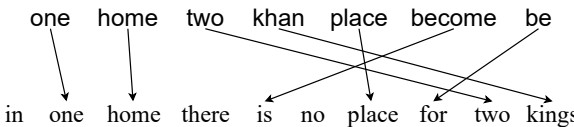

Figure 3: Example of SimAlign alignment between lexical glosses and an English translation (Tsez sentence).

Figure 3 displays an alignment computed with the `Match` method. Most alignments are trivial and associate identical units (e.g. one/'one') or morphologically related words (e.g. son/'sons'). Non-trivial (correct) links comprise (khan/'kings'),

[3]Assuming there are fewer lexical glosses than L2 words.

which is the best option as 'khan' does not occur in the translation. A less positive case is the (erroneous) alignment of be with 'for', which only exists because of the constraint of aligning every lexical gloss. Nevertheless, frequent lemmas such as 'be' will occur in multiple sentences, and their correct labels are often observed in other training sentences. We analyse these alignments in §5.4.

## 3.2 Implementing the hybrid CRF model

Following e.g. (Dyer et al., 2011; Lavergne et al., 2011), our implementation of the CRF model[4] heavily relies on weighted finite-state models and operations (Allauzen et al., 2007), which we use to represent the spaces of all possible and reference labellings on a per sentence basis, and to efficiently compute the expectations involved in the gradient, as well as to search for the optimal labellings and compute alignment and label posteriors.

Training is performed by optimising the penalised conditional log-likelihood with a variant of gradient descent (Rprop, (Riedmiller and Braun, 1993)), with $\ell_1$ regularisation to perform feature selection, associated with parameter value 0.5 that was set during preliminary experiments and kept fixed for producing all the results below.

Given the size of our training data (§4.1) and typical sentence lengths, training and decoding are very fast, even with hundreds of labels and millions of features. A full experiment for Lezgi takes about 20 minutes on a desktop machine; processing the larger Tsez dataset takes about 10 hours.

## 3.3 Observation and label features

Our implementation can handle a very large set of sparse feature functions $g_k()$, testing arbitrary properties of the input L1 sequence in conjunction with either isolated labels (unigram features) or pairs of labels (bigram features). Regarding L1, we distinguish between *orthographic features*, which test various properties of the morpheme string (its content, prefix, suffix, CV structure and length), and *positional features*, which give information about the position of the morpheme in a word; all these features can also test the properties of the surrounding morphemes (within the same word or in its neighbours). Note that a number of these features abstract away the orthographic content, a property we exploit in our multilingual model.

On the label side, feature functions test the gloss value $y$ and type $b$ (GRAM or LEX); for labels aligned with an L2 word, we additionally collect its PoS $p$[5] and its position $l$ in $z$, which acts as a distortion feature.[6] Such label features enable us to generalise alignment patterns for unknown L1 morphemes. More about features in Appendix A.

## 4 Experimental conditions

### 4.1 L1 languages

We consider five (out of seven) languages from the SIGMORPHON 2023 Shared Task on Interlinear Glossing (Ginn et al., 2023): Tsez (ddo), Gitksan (git), Lezgi (lez), Natugu (ntu; surprise language), and Uspanteko (usp; target translation in Spanish).[7] Table 2 gives general statistics about the associated datasets; a brief presentation of these languages is in Appendix B and in (Ginn et al., 2023).

| language | ddo | git | lez | ntu | usp |
| --- | --- | --- | --- | --- | --- |
| train | 3,558 | 31 | 701 | 791 | 9,774 |
| dev | 445 | 42 | 88 | 99 | 232 |
| test | 445 | 37 | 87 | 99 | 633 |

Table 2: Number of sentences for each language

### 4.2 Pre-processing L2 translations

L2 translations are lemmatised and PoS tagged with spaCy,[8] using the en_core_web_sm and es_core_news_sm pipelines for English and Spanish respectively. All lemmas in the translation are lowercased except for proper nouns.

### 4.3 Multilingual corpus

We also explore multilingual pre-training by leveraging a recent IGT corpus, IMTVault (Nordhoff and Krämer, 2022), which contains IGT examples from various publications in Language Science Press. In an attempt to capture cross-linguistic patterns, we only keep sentences with a well-defined language code with at least 30 sentences, since the languages we study range from 31 to thousands of sentences, leaving us 173 languages. The dataset

---

[5]For labels in $\mathcal{Y}_L(\boldsymbol{x})$, we associate the most frequently found PoS tag in the training dataset via alignment. As grammatical morphemes have no aligned target words, we use the generic label GRAM for all grammatical glosses.

[6]Non-aligned units (i.e. glosses from $\mathcal{Y}_L(\boldsymbol{x})$ and $\mathcal{Y}_G$) have dedicated positions ($-1$ and $-2$, respectively).

[7]We did not run our models on the Nyangbo (nyb) dataset, which does not include a translation tier.

[8]https://spacy.io/.

is shuffled and then split into 30K training, 2K development, and 2K test datasets.

## 4.4 SimAlign settings

Since the glosses and the translation are in the same language, we use the embeddings from the English BERT (`bert-base-uncased`) (Devlin et al., 2019) when the L2 language is English and mBERT ('`bert-base-multilingual-uncased`') for Spanish (for Uspanteko). We stress here that our model is compatible with several target languages, SimAlign being an off-the-shelf multilingual (neural) aligner.

Our preliminary experiments showed that the embeddings from the 0-th layer yielded the best alignments, especially compared to the 8-th layer, which seems to work best in most alignment tasks. A plausible explanation is that contextualised embeddings are unnecessary here because lexical glosses do not constitute a standard English sentence (for instance, they do not contain stop words, and their word order reflects the L1 word order).

## 4.5 Evaluation metrics

We use the official evaluation metrics from the Shared Task: morpheme accuracy, word accuracy, BLEU, and precision, recall, and F1-score computed separately for grammatical (Gram) and lexical (Lex) glosses. We report the results of our best system with all metrics in Appendix.

## 4.6 Baselines

Below, we consider three baseline approaches which handle glossing as a sequence-labelling task:

- `maj`: a dictionary-based approach, which assigns the *majority* label (grammatical and lexical) seen in the training dataset to a source morpheme and fails for out-of-vocabulary morphemes;
- `CRF+maj`: a hybrid model relying on a CRF to predict grammatical glosses and a unified lexical label (as in (Moeller and Hulden, 2018; Barriga Martínez et al., 2021)). Known lexical morphemes are then assigned a lexical label according to the `maj` model;
- `BASE_ST`: is the Transformer-based baseline developed by the SIGMORPHON Shared Task organisers and detailed in Ginn (2023).[9]

[9]https://github.com/sigmorphon/2023glossingST/tree/main/baseline.

# 5 Experiments

## 5.1 Results

Table 3 reports the scores of the baselines from §4.6, as well as the best results in the Shared Task on Automatic Glossing (BEST_ST)[10] and the results of variants of our system on the official testsets. We only report below the word- and morpheme-level (overall) accuracy, which are the two official metrics of the Shared Task.[11]

| model | ddo | git | lez | ntu | usp |
|---|---|---|---|---|---|
| maj | 65.3 | 28.1 | 81.2 | 81.5 | 72.8 |
| CRF+maj | - | 29.4 | 84.9 | 88.1 | 76.2 |
| BASE_ST | 75.7 | 16.4 | 34.5 | 41.1 | 76.6 |
| BEST_ST | **85.8** | 31.5 | **85.4** | **89.3** | 78.5 |
| CRF (S1) | 36.2 | 25.5 | 65.3 | 59.0 | 53.2 |
| CRF (S2) | 51.5 | 29.9 | 52.8 | 65.0 | 63.3 |
| CRF (S3) | 85.6 | **33.6** | 82.8 | 89.1 | **78.9** |
| maj | 79.1 | 51.2 | 85.8 | 87.1 | 79.5 |
| CRF+maj | - | 51.1 | **88.3** | 92.3 | 82.5 |
| BASE_ST | 85.3 | 25.3 | 51.8 | 49.0 | 82.5 |
| BEST_ST | **92.0** | **52.4** | 87.6 | **92.8** | **84.5** |
| CRF (S1) | 60.1 | 47.8 | 73.5 | 73.0 | 65.5 |
| CRF (S2) | 70.2 | 45.3 | 65.0 | 76.8 | 73.3 |
| CRF (S3) | 91.9 | **52.4** | 87.0 | **92.8** | 84.4 |

Table 3: Accuracy (overall) at the word (top) and morpheme (bottom) levels for baseline systems and three variants of the hybrid CRF model on the *test* dataset. Best scores in each language & metrics are in **bold**.[12]

A first observation is that system (S3), which effectively combines the information available in a dictionary and obtained via alignment in an integrated fashion (see §2.2) greatly outperforms (S1) (only dictionary) and (S2) (only alignment), obtaining the best performance among our three variants. (S3) is also consistently better than all the baselines, with larger gaps when few training sentences are available (e.g. Gitksan or Lezgi). In comparison, the BERT-based baseline suffers a large performance drop in very low-data settings, as also reported in (Ginn, 2023). Our CRF model also achieves competitive results compared to the best system submitted to the Shared Task, especially for the word-level scores. These scores confirm

[10]https://github.com/sigmorphon/2023glossingST/blob/main/results.md.

[11]Full results of CRF (S3) in Appendix D.

[12]CRF+maj was not computed on Tsez because the number of features is too prohibitive. For our systems, CRF (S1) and CRF (S2) are the lower bounds of our results.

that decent to good accuracy numbers can be obtained based on some hundreds of training sentences. Note, however, that annotated datasets of that size are not so easily found: in the IMTVault (Nordhoff and Krämer, 2022), only 16 languages have more than 700 sentences, which is about the size of the Lezgi and the Natugu corpora.

## 5.2 Handling unknown morphemes

Leveraging the translation in glossing opens the way to better handle morphemes that were unseen in training. Table 4 displays some statistics about unknown morphemes in test datasets. For most languages, they are quite rare, representing solely around or below 10% of all lexical glosses in the test set, with Gitksan a notable outlier (Ginn et al., 2023). Among those unseen morphemes, a significant proportion (from a third in Tsez up to 70% in Gitksan) of the reference lexical gloss is not even present in the translation[13] (cf. 'not in L2' in Table 4). Taking this into account nuances the seemingly low accuracy for unknown lexical morphemes: in Uspanteko, for instance, the system reaches an accuracy of 29.3 when the best achievable score is about 35. To have a more optimistic view of our prediction, we 'approximate' *lexical* glosses with lemmas from the translation, using automatic alignments (e.g., king instead of khan in Figure 3). By evaluating the unknown morphemes with their reachable labels[14] (cf. 'align. accuracy' line), we get higher scores, such as 40.4 in Natugu.

| language | ddo | git | lez | ntu | usp |
|---|---|---|---|---|---|
| number | 44 | 200 | 64 | 47 | 181 |
| proportion (%) | 1.02 | 74.9 | 9.65 | 6.16 | 10.9 |
| not in L2 | 15 | 143 | 42 | 24 | 117 |
| gold accuracy | 18.2 | 12.0 | 4.7 | 29.8 | 29.3 |
| align. accuracy | 15.9 | 18.0 | 10.9 | 40.4 | 56.9 |

Table 4: Statistics about unknown *lexical* morphemes in testsets. We report the morpheme-level accuracy.

## 5.3 Data efficiency

An important property of our approach seems to be its data efficiency. To better document this property, we report in Table 5 the morpheme-level accuracy obtained with increasingly large training data of

size (50, 200, 700, 1,000, 2,000, full) in Tsez. With 200 examples already, our model does much better than the simple baseline maj and delivers usable outputs. (s3) also has a faster improvement rate than the baseline for small training datasets (e.g. almost +7 points between 200 and 700 sentences), while maj increases by only +4 points. The return of increasing the dataset size above 1,000 is, in comparison, much smaller. While the exact numbers are likely to vary depending on the language and the linguistic variety of the material collected on the field, they suggest that ML techniques could be used from the onset of the annotation process.

| train | 50 | 200 | 700 | 1,000 | 2,000 | full |
|---|---|---|---|---|---|---|
| maj | 61.0 | 72.4 | 76.7 | 77.6 | 78.7 | 79.1 |
| CRF (s3) | 66.9 | 80.6 | 87.5 | 89.2 | 90.7 | 91.9 |

Table 5: Morpheme-level accuracy with increasing training data size in Tsez for two systems.

## 5.4 Analysis of automatic alignments

Our approach relies on automatic alignment computed with SimAlign to supervise the learning process. When using the Match method, (almost) all lexical glosses are aligned with a word in the translation (cf. footnote 3). We cannot evaluate the overall alignment quality, as reference alignments are unobserved. However, we measure in Table 6 the proportion of *exact* matches between the reference gloss and the lemma of the aligned word.

| model | ddo | git | lez | ntu | usp |
|---|---|---|---|---|---|
| exact match | 51.3 | 43.2 | 51.2 | 60.6 | 47.8 |

Table 6: Proportion of *exact* matches between the reference gloss and the lemma of the aligned word with SimAlign for the training datasets.

Overall, around half of our alignments are trivial and hence sure, which is more or less in line with the proportions found by Georgi (2016), albeit lower due to the marked linguistic difference between L1 and L2 in our case. These seemingly low scores have to be nuanced by two facts. First, they are a lower bound to evaluate alignment quality since synonyms (such as khan/king) are counted as wrong alignments. In some cases, inflected forms are also used as a gloss (e.g., dijo/decir in Uspanteko). Second, the alignment-induced glosses are not used as is in our experiments: they supplement

---

[13]We studied the exact match with any lemmas in the translation; composed glosses are hence always considered as absent.

[14]Whenever possible; if the lexical gloss has no automatic alignment, we keep the reference gloss.

the output label with their PoS tag and their position in the L2 sentence. This means that even non-exact matches can yield useful features.

**Removing L2 stop words**  We carried out a complementary experiment where we filtered stop words in the L2 translation in Gitksan to remove a potential source of error in the alignments. The number of unaligned lexical glosses (§3.1) thus increases, which generally means a reduced noise in the alignment for the Match method. Yet, using these better alignments and reduced label sets in training and inference yields mixed results: $+1$ point in word accuracy, $-2$ points in morpheme accuracy.

### 5.5 Multilingual pre-training

Cross-lingual transfer techniques via multilingual pre-training (Conneau et al., 2020) are the cornerstone of multilingual NLP and have been used in multiple contexts and tasks, including morphological analysis for low-resource languages (Anastasopoulos and Neubig, 2019). In this section, we apply the same idea to evaluate how well such techniques can help in our context. We train the model to predict the nature of the gloss (grammatical or lexical) with multilingual features (see Appendix A): for a given morpheme, its position in the word, its length in characters, its CV skeleton,[15] and the number of morphemes in the word. Using IMTVault (§4.3) for this task, the model reaches around 80 of accuracy.

| model | ddo | git | lez | ntu | usp |
|---|---|---|---|---|---|
| CRF (S3) | 85.6 | 33.6 | 82.8 | 89.1 | 78.9 |
| + IMT | 85.3 | 33.1 | 83.4 | 89.0 | 79.0 |
| CRF (S3) | 91.9 | 52.4 | 87.0 | 92.8 | 84.4 |
| + IMT | 91.8 | 52.8 | 87.1 | 92.5 | 84.5 |

Table 7: Accuracy (overall) at the word (top) and morpheme (bottom) levels for the model without and with multilingual pre-training on the *test* dataset.

We use these pre-trained weights to initialise the multilingual features in our (S3) system. To help feature selection, we notably reduce the value of $\ell_1$ to $0.4$. Pre-training results (+ IMT) are in Table 7. We note that pre-training has a negligible effect, except in some metrics, such as in Lezgi for the word level. We observed that the most important weights in the pre-trained model correspond

[15]We identify consonants and vowels based on orthography.

to delexicalised pattern features that are relevant in IMTVault but not present at all in our datasets.

**Cross-lingual study**  Besides, since in our studied languages, Tsez and Lezgi belong to the same language family, we reiterate this methodology but with the Tsez dataset as a pre-training source to predict Lezgi glosses. This kinship is also explored through successful cross-lingual transfer in (Zhao et al., 2020). We obtain 83.3 and 87.1 for word and morpheme accuracy, which is very close to the performance with (and without) IMTVault despite containing fewer sentences.

**Very low resource scenarios**  A final experiment focuses on a very low training data scenario. Save the Gitksan corpus, the test languages already represent hundreds of annotated sentences. This contrasts with actual data conditions: in IMTVault, for instance, only 16 languages have equivalent or more sentences than Lezgi (the second-lowest language in terms of training sentences in our study; cf. §5.1). We thus focus here on (simulated) very low-resource data settings by considering only 50 sentences[16] selected from the training data.

| model | ddo | lez | ntu | usp |
|---|---|---|---|---|
| CRF (S3) | 47.6 | 53.7 | 64.9 | 45.9 |
| + IMT | 48.0 | 54.0 | 65.6 | 48.3 |
| CRF (S3) | 66.9 | 63.3 | 74.7 | 57.5 |
| + IMT | 67.3 | 63.4 | 75.3 | 59.2 |

Table 8: Accuracy at the word (top) and morpheme (bottom) levels for (S3) without and with multilingual pre-training on the *test* dataset with 50 training sentences.

Table 8 reports the results obtained with this configuration. We observe that using the multilingual pre-training helps for all languages to a more noticeable extent than before. These three experiments confirm the potential of multilingual transfer for this task, which can help improve performance in very low-resource scenarios. Contrarily, when hundreds of sentences are available, pre-training delexicalised features proves ineffective, sometimes even detrimental.

### 5.6 Feature analysis

One positive point in using models based on CRFs relies on access to features and their correspond-

[16]Filtering IMTVault with this threshold would lead to 124 languages kept.

ing learnt weights. We report in Appendix E ten features with the largest weight in Natugu.

Among the top 1% of active features in Natugu in terms of weight, we find two features testing the gloss type $b$ with the morpheme length: (LEX, 6+) and (LEX, 5). These indicate that longer morphemes are likely to have a lexical label. Such an analysis can also be relevant to weights learnt through multilingual pre-training. For instance, (LEX, 5) is among the top 10% features on the IMT-Vault dataset, suggesting a cross-lingual tendency of longer morphemes being lexical.

## 6 Related work

**Language documentation** With the ever-pressing need to collect and annotate linguistic resources for endangered languages, the field of computational language documentation is quickly developing, as acknowledged by the ComputEL workshop.[17] Regarding annotation tools, recent research has focused on all the steps of language documentation, from speech segmentation and transcription to automatic word and morpheme splitting to automatic interlinear glossing.

(Xia and Lewis, 2007; Georgi et al., 2012, 2013) use, as we do, interlinear glosses to align morphs and translations, then to project syntactic parses from L2 back to L1, a technique pioneered by Hwa et al. (2005), or to extract grammatical patterns (Bender et al., 2014). Through these studies, a large multilingual database of IMGs was collected from linguistic papers, curated, and enriched with additional annotation layers (Lewis and Xia, 2010; Xia et al., 2014) for more than 1,400 languages. (Georgi, 2016) notably discusses alignment strategies in ODIN and trains a multilingual alignment model between the gloss and translation layers - in our work, we extend multilingual training to the full annotation process. Another massive multilingual source of glosses is IMTVault, described in (Nordhoff and Krämer, 2022), studied in §5.5.

**Automatic glossing** Automatic glossing was first studied in (Palmer et al., 2009; Baldridge and Palmer, 2009), where active learning was used to incrementally update an underlying tagging system focusing mainly on grammatical morphemes (lexical items are tagged with their PoS). (Samardžić et al., 2015) added to this an extra layer aimed to annotate the missing lexical tags, yielding a system that resembles our CRF+maj baseline.

---

[17] https://computel-workshop.org/.

Both (Moeller and Hulden, 2018) and (McMillan-Major, 2020) rely on CRFs, the latter study being closer to our approach as it tries to combine post-hoc the output of two CRF models operating respectively on the L1 and L2 tier, where our system introduces an integrated end-to-end architecture. Zhao et al. (2020) develop an architecture inspired by multi-source neural translation models, where one source is the L1 sequence, and the other the L2 translation. They experiment with Arapaho, Lezgi, and Tsez, while also applying some sort of cross-lingual transfer learning. The recent SIGMORPHON exercise (Ginn et al., 2023) is a first attempt to standardise benchmarks and task settings and shows that morpheme-level accuracy in the high 80s can be obtained for most languages considered.

**Latent variable CRFs models** Extended CRF models were proposed and used in many studies, including latent variables to represent, e.g. hidden segmentation as in (Peng et al., 2004) or hidden syntactic labels (Petrov and Klein, 2007). Closer to our work, (Blunsom et al., 2008; Lavergne et al., 2011) use latent structures to train discriminative statistical machine translation systems. Other relevant work on unsupervised discriminative alignment are in (Berg-Kirkpatrick et al., 2010; Dyer et al., 2011), while Niehues and Vogel (2008) use a supervised symmetric bi-dimensional word alignment model.

## 7 Conclusion

This paper presented a hybrid CRF model for the automatic interlinear glossing task, which was specifically designed and tailored to work well in low data conditions and to effectively address issues due to out-of-vocabulary morphemes. We presented our main approach, which relies on analysing the translation tier, and discussed our main implementation choices. In experiments with five low-resource languages, we obtained accuracy scores that match or even outperform those of very strong baselines, confirming that accuracy values in the 80s or above could be obtained with a few hundred training examples. Using a large multilingual gloss database, we finally studied the possibility of performing cross-lingual transfer for this task.

There are various ways to continue this work and improve these results, such as removing the noise introduced via erroneous alignments links—either by marginalising over the 'origin' variable, by filtering unlikely alignments based on link poste-

rior values, or by also trying to generate alignment links for function words (Georgi, 2016; McMillan-Major, 2020). Our initial experiments along these lines suggest that this may not be the most promising direction. We may also introduce powerful neural representations for L1 languages; while these were usually available for a restricted number of languages, recent works have shown that even low-resource languages could benefit from these techniques (Wang et al., 2022; Adebara et al., 2023).

## Limitations

The main limitation comes from the small set of languages (and corresponding language family) studied in this work. In general, texts annotated with Interlinear Morphological Gloss are scarcely available due to the time and expertise needed to annotate sentences with glosses. However, corpora such as IMTVault (Nordhoff and Krämer, 2022) or ODIN (Lewis and Xia, 2010) or languages such as Arapaho (39,501 training sentences in the Shared Task) pave the way for further experiments.

Moreover, another shortcoming of our work stems from the fact that we do not use neural models in our work, while, for instance, the best submission to the Shared Task relies on such architectures. In this sense, we have yet to compare the scalability of our performance to larger data. Still, one of our main focuses was to tackle glossing in very low-resource situations for early steps of language documentation and to study how to handle previously unseen morphemes at the inference step.

## Acknowledgements

We thank the anonymous reviewers and meta-reviewer for their comments and suggestions. This work was partly funded by French ANR and German DFG under grant ANR-19-CE38-0015 (CLD 2025). The authors warmly thank Thomas Lavergne for his help and assistance regarding the configuration and exploitation of Lost.

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

## A  Model features

The input side $x$ is the L1 morpheme sequence; from each $x_t$, from which we also deduce the following features: its position $t$ within the word coded as a numerical value (from $0$ to $n$) for complex words, or as 'F' for free morphemes, its length $l$ in characters, its 3-char prefix and suffix ($d$ and $e$ respectively), the number of morphemes in the word, and its CV skeleton $dl$ (Consonant and Vowels) based on the orthography of the morpheme. Figure 4 displays an example of input and the associated features, while Table 9 illustrates the output label format according to the origin of the gloss.

With all these inputs to predict the output labels, we compute unigram and bigram feature functions, detailed in Table 13.

Besides, we add features that test properties between the source and translation: a copy feature to

| i | input source morph. $m$ | position (in word) $t$ | length $l$ | first 3 letters $d$ | last 3 letters $e$ | copy src $cs$ | position src $ps$ | CV skeleton $dl$ | reference gloss $y$ | GRAM or LEX $b$ | PoS tag $p$ | copy trg $ct$ | position trg $pt$ |
|---|---|---|---|---|---|---|---|---|---|---|---|---|---|
| 0 | nesi | 0 | 4 | nes | esi | 0 | 1/4 | CVCV | he.OBL | LEX | PRON | 0 | 1/4 |
| 1 | s | 1 | 1 | s | s | 0 | 1/4 | C | GEN1 | GRAM | GRAM | -1 | -2 |
| 2 | ʕono | F | 5 | ʕo | ono | 0 | 2/4 | CCVCV | three | LEX | NUM | 0 | 3/4 |
| 3 | uži | F | 3 | uži | uži | 0 | 2/4 | VCV | son | LEX | NOUN | 0 | 4/4 |
| 4 | zow | 0 | 3 | zow | zow | 0 | 3/4 | CVC | be.NPRS | LEX | VERB | 0 | 2/4 |
| 5 | n | 1 | 1 | n | n | 0 | 4/4 | C | PST.UNW | GRAM | GRAM | -1 | -2 |

Figure 4: Example of input, outputs, and associated features to Lost for the Tsez reference sentence of Figure 1.

| Origin | $y$ | $b$ | $p$ | $l$ |
|---|---|---|---|---|
| $\mathcal{Y}_G$ | GEN1 | GRAM | GRAM | - |
| $\mathcal{Y}_L(\boldsymbol{x})$ | king | LEX | NOUN | - |
| $z_{10}$ | khan | LEX | NOUN | 10 |
| $y_t, o_t = 10$ | khan | LEX | NOUN | 10 |

Table 9: Example of output features extracted from each label set, using the example of Figure 3. The reference $(y_t, o_t)$ is the training supervision.

handle glosses that literally appear in the L2 sentence (namely, proper nouns) and a distortion feature which tests the difference in relative positions between the source morpheme and the (possible) lexical label, whenever $a_t > 0$.

## B  Brief language presentation

We describe below the studied languages.

- Tsez (ddo) is a Nakh-Daghestanian language spoken in the Republic of Dagestan in Russia.

- Gitksan (git) is a Tsimshian language spoken on the western coast of Canada (British Columbia).

- Lezgi (lez) is a Nakh-Daghestanian language spoken in the Republic of Dagestan in Russia and in Azerbaijan.

- Natugu (ntu) is an Austronesian language spoken in the Solomon Islands.

- Uspanteko (usp) is a Mayan language spoken in Guatemala.

According to Ethnologue (Eberhard et al., 2023), the two Nakh-Daghestanian languages have between 10K to 1M speakers, while the other three have fewer than 10K users.

## C  Number of active features

Table 10 presents the number of active features (in thousands) selected among all features (in mil-

lions) for s3. We note here that thanks to the $l_1$-regularisation term, most feature weights are set to 0 and less than 1% of the features are actually retained.

| | ddo | git | lez | ntu | usp |
|---|---|---|---|---|---|
| Active s3 | 196k | 4k | 51k | 73k | 151k |
| Total | 188M | 1M | 26M | 44M | 29M |

Table 10: Number of selected features among all computed features for the s3 system in each language.

## D  Full results

Table 11 displays the results of the s3 system with all metrics presented in §4.5. Two scores of accuracy are computed for both morpheme and word levels: an overall (Ovr) value and a sentence-averaged value (Avg).

## E  Example of learnt features

Table 12 displays 10 features with the largest weight in the s3 system in Natugu. Here, we ignore trivial features for numbers or punctuation signs that also have high weight values. The ID of the feature refers to Table 13.

| | Morpheme Acc. | Word Acc. | BLEU (Morpheme) | Lex | Gram |
|---|---|---|---|---|---|
| | Ovr: 91.9 | Ovr: 85.6 | | P: 91.7 | P: 92.1 |
| Tsez (ddo) | | | 81.3 | R: 90.7 | R: 93.0 |
| | Avg: 91.9 | Avg: 85.6 | | F1: 91.2 | F1: 92.5 |
| | Ovr: 52.4 | Ovr: 33.6 | | P: 23.0 | P: 76.7 |
| Gitksan (git) | | | 19.2 | R: 27.0 | R: 68.4 |
| | Avg: 53.1 | Avg: 33.1 | | F1: 24.8 | F1: 72.3 |
| | Ovr: 87.0 | Ovr: 82.8 | | P: 87.2 | P: 86.7 |
| Lezgi (lez) | | | 73.1 | R: 87.5 | R: 86.2 |
| | Avg: 86.4 | Avg: 82.0 | | F1: 87.3 | F1: 86.5 |
| | Ovr: 92.8 | Ovr: 89.1 | | P: 90.1 | P: 95.5 |
| Natugu (ntu) | | | 82.8 | R: 89.8 | R: 95.8 |
| | Avg: 92.6 | Avg: 88.7 | | F1: 89.9 | F1: 95.6 |
| | Ovr: 84.4 | Ovr: 78.9 | | P: 78.1 | P: 89.9 |
| Uspanteko (usp) | | | 69.4 | R: 76.3 | R: 91.8 |
| | Avg: 79.3 | Avg: 74.6 | | F1: 77.2 | F1: 90.8 |

Table 11: Full results with our S3 system on the IGT Shared Task test dataset.

| ID | Feature | Weight |
|---|---|---|
| 25 | be ∧ 1/4 ∧ ngi ∧ 3/4 | 2.91 |
| 9 | PREP ∧ NMLZ1 ∧ mz | 2.49 |
| 25 | market ∧ 4/4 ∧ maket ∧ 2/4 | 2.49 |
| 8 | body ∧ rtq | 2.36 |
| 25 | tree ∧ 3/4 ∧ nc ∧ 1/4 | 2.29 |
| 7 | SUBR ∧ kx | 2.28 |
| 3 | SUBR ∧ kx | 2.28 |
| 25 | catechist ∧ 4/4 ∧ katkis ∧ 2/4 | 2.28 |
| 8 | take ∧ twz | 2.28 |
| 25 | eat ∧ 3/4 ∧ mu ∧ 4/4 | 2.28 |

Table 12: 10 features with the largest weight in Natugu.

| ID | Feature | Test | Example (cf. Figure 4 $i = 5$) |
|---|---|---|---|
| 1 | uni-gloss | $\mathbb{1}(g_i = g)$ | PST.UNW |
| 2 | bi-gloss | $\mathbb{1}(g_i = g) \wedge \mathbb{1}(g_{i-1} = g')$ | (be.NPRS, PST.UNW) |
| 3 | uni-gloss-morph | $\mathbb{1}(g_i = g) \wedge \mathbb{1}(m_i = m)$ | (PST.UNW, n) |
| 4 | uni-gloss-bi-morph | $\mathbb{1}(g_i = g) \wedge \mathbb{1}(m_{i-1} = m') \wedge \mathbb{1}(m_i = m)$ | (PST.UNW, zow, n) |
| 5 | uni-gloss-position | $\mathbb{1}(g_i = g) \wedge \mathbb{1}(t_i = t)$ | (PST.UNW, 1) |
| 6 | uni-gloss-length | $\mathbb{1}(g_i = g) \wedge \mathbb{1}(l_i = l)$ | (PST.UNW, 1) |
| 7 | uni-gloss-start | $\mathbb{1}(g_i = g) \wedge \mathbb{1}(d_i = d)$ | (PST.UNW, n) |
| 8 | uni-gloss-end | $\mathbb{1}(g_i = g) \wedge \mathbb{1}(e_i = e)$ | (PST.UNW, n) |
| 9 | bi-gloss-morph | $\mathbb{1}(g_i = g) \wedge \mathbb{1}(g_{i-1} = g') \wedge \mathbb{1}(m_i = m)$ | (be.NPRS, PST.UNW, n) |
| 10 | *uni/bi-bin | $\mathbb{1}(b_i = b) \, (\wedge\mathbb{1}(b_{i-1} = b'))$ | GRAM ((LEX, GRAM)) |
| 11 | uni/bi-pos | $\mathbb{1}(p_i = p) \, (\wedge\mathbb{1}(p_{i-1} = p'))$ | GRAM ((VERB, GRAM)) |
| 12 | uni-bin-morph | $\mathbb{1}(b_i = b) \wedge \mathbb{1}(m_i = m)$ | (GRAM, n) |
| 13 | *uni-bin-position/length | $\mathbb{1}(b_i = b) \wedge \mathbb{1}(t_i = t)/\mathbb{1}(l_i = l)$ | (GRAM, 1) / (GRAM, 1) |
| 14 | uni-bin-start/end | $\mathbb{1}(b_i = b) \wedge \mathbb{1}(d_i = d)/\mathbb{1}(e_i = e)$ | (GRAM, n) / (GRAM, n) |
| 15 | *uni-bin-bi-position | $\mathbb{1}(b_i = b) \wedge \mathbb{1}(t_i = t) \wedge \mathbb{1}(t_{i-1} = t')$ | (GRAM, 0, 1) |
| 16 | bi-bin-gloss | $\mathbb{1}(g_i = g) \wedge \mathbb{1}(b_{i-1} = b')$ | (LEX, PST.UNW) |
| 17 | bi-gloss-bin | $\mathbb{1}(b_i = b) \wedge \mathbb{1}(g_{i-1} = g')$ | (be.NPRS, GRAM) |
| 18 | uni-pos-morph | $\mathbb{1}(p_i = p) \wedge \mathbb{1}(m_i = m)$ | (GRAM, n) |
| 19 | bi-pos-gloss | $\mathbb{1}(g_i = g) \wedge \mathbb{1}(p_{i-1} = p')$ | (VERB, PST.UNW) |
| 20 | bi-gloss-pos | $\mathbb{1}(p_i = p) \wedge \mathbb{1}(g_{i-1} = g')$ | (be.NPRS, GRAM) |
| 21 | uni-pos-start/end | $\mathbb{1}(p_i = p) \wedge \mathbb{1}(d_i = d)/\mathbb{1}(e_i = e)$ | (GRAM, n) / (GRAM, n) |
| 22 | uni-copy-trg | $\mathbb{1}(ct_i = ct)$ | -1 |
| 23 | uni-copy-trg-src | $\mathbb{1}(ct_i = ct) \wedge \mathbb{1}(cs_i = cs)$ | (-1, 0) |
| 24 | uni-posi-ts | $\mathbb{1}(pt_i = pt) \wedge \mathbb{1}(ps_i = ps)$ | (-2, 4/4) |
| 25 | uni-gloss-morph-pts | $\mathbb{1}(g_i = g) \wedge \mathbb{1}(pt_i = pt)$ $\wedge\mathbb{1}(m_i = m) \wedge \mathbb{1}(ps_i = ps)$ | (PST.UNW, -2, n, 4/4) |
| 26 | uni-gloss-delex | $\mathbb{1}(g_i = g) \wedge \mathbb{1}(dl_i = dl)$ | (PST.UNW, C) |
| 27 | *uni-bin-delex | $\mathbb{1}(b_i = b) \wedge \mathbb{1}(dl_i = dl)$ | (GRAM, C) |
| 28 | uni-pos-delex | $\mathbb{1}(p_i = p) \wedge \mathbb{1}(dl_i = dl)$ | (GRAM, C) |
| 29 | *uni-bin-bi-delex | $\mathbb{1}(b_i = b) \wedge \mathbb{1}(dl_{i-1} = dl') \wedge \mathbb{1}(dl_i = dl)$ | (GRAM, CVC, C) |
| 30 | *bi-bin-bi-delex | $\mathbb{1}(b_{i-1} = b') \wedge \mathbb{1}(b_i = b)$ $\wedge\mathbb{1}(dl_{i-1} = dl') \wedge \mathbb{1}(dl_i = dl)$ | (LEX, PST.UNW, CVC, C) |
| 31 | *uni-bin-rel-morph-position | $\mathbb{1}(b_i = b) \wedge \mathbb{1}(t_i = t) \wedge \mathbb{1}(ml_i = ml)$ | (GRAM, 1, 2) |

Table 13: Unigram and bigram features for our submissions: features about the main gloss label on top, those involving the two other general outputs, and other additional features at the bottom. Star-marked feature names indicate *multilingual* features also used for pre-training.