# OpenReview forum: "Towards Multilingual Interlinear Morphological Glossing"
_EMNLP/2023/Conference — EMNLP 2023 Findings_

### Official Review · Reviewer_v4Qx · 2023-07-26

**Soundness:** 2

**Excitement:**

3: Ambivalent: It has merits (e.g., it reports state-of-the-art results, the idea is nice), but there are key weaknesses (e.g., it describes incremental work), and it can significantly benefit from another round of revision. However, I won't object to accepting it if my co-reviewers champion it.

**Paper Topic And Main Contributions:**

This work focuses on automatic interlinear morphological glossing. An approach by CRF is proposed and several low-resource languages are taken into experiments and discussions.

**Reasons To Accept:**

- Personally, I like the topic.

**Reasons To Reject:**

- The technical issues are not well presented.
- For heavily inflexional or agglutinative languages, the possible labels may be huge. Is the CRF a good option?
- The whole work is organized in an engineering-oriented manner, while the topic should be linguistic-oriented.

**Reproducibility:**

3: Could reproduce the results with some difficulty. The settings of parameters are underspecified or subjectively determined; the training/evaluation data are not widely available.

**Reviewer Confidence:**

3: Pretty sure, but there's a chance I missed something. Although I have a good feel for this area in general, I did not carefully check the paper's details, e.g., the math, experimental design, or novelty.

---

> ### Author Rebuttal · Authors · 2023-08-26
>
> Thank you for your review.
>
> - We strive to improve the clarity of our work. It would be of tremendous help if you could point us out some technical issues where you feel more detail would be helpful.
>
> - For the languages we studied, which contain very morphologically-rich languages such as Lezgi, Natugu or Tsez, the set of grammatical labels is around a few hundred. This size of labels is still manageable with linear-chains CRFs (see e.g. (Mueller et al., 2013; Lavergne and Yvon, 2017), which develop CRFs with very large sets of labels (up to several hundreds).
>
> - EMNLP is about empirical methods in NLP, which is what we report; so it is not immediately clear to us where a more linguistic-oriented presentation would be needed. It would greatly help us to improve the paper if the reviewer could expand their comment.
>
> We finally note an issue with reproducibility: as we will release the code (footnote 4) and only experiment with publicly available data, it is not clear which details are missing for reproducibility.

---

### Official Review · Reviewer_fF76 · 2023-07-27

**Soundness:** 4

**Excitement:**

4: Strong: This paper deepens the understanding of some phenomenon or lowers the barriers to an existing research direction.

**Paper Topic And Main Contributions:**

The paper proposes an approach for interlinear morphological glossing (IMG) of a sentence that also makes use of a translation of the sentence. This version of the IMG problem aims at, given the morphemes of a sentence in a low-resource language and its translation in a high-level language, determining the morphological analysis/glossing of the sentence. The paper views the problem as a hybrid problem formed of two parts which are sequence labeling and sequence alignment. A CRF model is used and it is adapted to the problem as taking two input parameters and outputting the gloss sequence. Experiments were conducted on five low-resource languages by using English as the high-resource language. Ablation studies were performed.

**Questions For The Authors:**

- Why did you prefer CRF rather than a neural model? Why didn't you compare the results with a neural approach?
- In Section 1, by "open vocabulary problem", do you mean that the number of possible combinations of morphological features (OBL, GEN1, etc.) is too large? In fact, I think that this is not exactly an open vocabulary issue. Although the number of combinations may be high, it is not impractical to consider all (as opposed to, e.g., the number of derived words in a morphologically rich language, which is nearly infinite due to properties like recursion).
- What do you mean exactly by "such cases are considered as lexical glosses" in footnote 1? It should be stated more clearly.
- At the end of Section 2.1, you mention two problems and also the labels GRAM and LEX. I couldn't see what these problems are for and what these labels are exactly.
- The part "In our model, ... because of (ii)" is not clear, I cannot catch what you mean. Please rewrite more clearly.

**Reasons To Accept:**

The paper focuses on a less-studied problem and shows the usefulness of the proposed approach on several very low-resource languages. As a novel contribution, translation of the given sentence is used and the open vocabulary problem (a large number of morphological feature combinations in the output) is addressed. A detailed set of ablation experiments were given including very low-resource scenarios and a feature analysis.

**Reasons To Reject:**

- It is not clear why the authors preferred a CRF model which, as the authors stated, is a good model in the "pre-neural era" and why they didn't compare the results with an LSTM- or transformer-based neural model. This seems as a weakness of the paper.
- Although the paper is generally well-written, there are some parts that are difficult to understand and that makes following the paper difficult Some of them are mentioned below.

**Reproducibility:**

2: Would be hard pressed to reproduce the results. The contribution depends on data that are simply not available outside the author's institution or consortium; not enough details are provided.

**Reviewer Confidence:**

3: Pretty sure, but there's a chance I missed something. Although I have a good feel for this area in general, I did not carefully check the paper's details, e.g., the math, experimental design, or novelty.

**Typos Grammar Style And Presentation Improvements:**

one-to-one correspondence the -> one-to-one correspondence with the
the feature function Gk only test -> the feature function Gk only tests
L2 translation are
CRF model the test dataset -> CRF model on the test dataset

---

> ### Author Rebuttal · Authors · 2023-08-26
>
> Thank you for your comments.
>
> Q1: The reason behind the choice of CRF stems from previous studies (Moeller and Hulden, 2018; Barriga Martínez et al., 2021), where CRF outperformed neural methods (RNN, LSTM, and biLSTM), mainly due the latter needing more training data, which proves to be a challenge in the case of very low-resource languages. The data efficiency of our model is discussed in section 5.3 (line 402).
> That being said, the Shared Task on automatic glossing provides several competitive neural alternatives evaluated on the same data. One of them, based on Transformers (RoBERTa), is already reported as BASE_ST in our paper (see lines 349-351); several others are documented in (Ginn et al, 2023). We will stress this point again in the final version.
>
> Q2: The open vocabulary problem in this paper concerns lexical glosses. As you rightfully point out, the set of grammatical labels is fixed for a given language, but the set of lexical labels is unbounded. Lexical labels seen in the training data are often not sufficient to cover all the lexical labels occurring in the evaluation data (out-of-vocabulary issue), which is why we use the translation to identify new possible labels (lines 49-54). We will emphasise this issue to make it clearer in the final version.
>
> Q3: By 'such cases are considered as lexical glosses' in footnote 1, we wanted to state our decision on handling 'compound' glosses such as he.OBL (lexical for he, and grammatical for OBL). Because of its lexical part, we considered these kinds of glosses as merely lexical, and no specific treatment has been done. This way, we just consider two types of glosses. We will explain this in more detail in the final version of the paper.
>
> Q4: The end of section 2.1 will be clarified in the final version. The two sets of labels denote simplified versions of the task, where we for instance just predict the gloss category (GRAM or LEX), but not its value.
>
> Q5: The discussion at the end of page 1 expresses that our model brings together two approaches. On the one hand, the set of grammatical glosses (i) and labels from the dictionary (iii) give us a fixed set, as in a standard sequence labelling task, while, on the other hand, predicting lemmas from the target sentence (ii), through their indices, makes it closer to an alignment problem. We will better explain this in the final version of the paper.
>
> We finally note an issue with reproducibility: as we will release the code (footnote 4) and only experiment with publicly available data, it is not clear which details are missing for reproducibility.
>
> Thank you for pointing out typos.

---

### Official Review · Reviewer_H7gr · 2023-08-04

**Soundness:** 4

**Excitement:**

3: Ambivalent: It has merits (e.g., it reports state-of-the-art results, the idea is nice), but there are key weaknesses (e.g., it describes incremental work), and it can significantly benefit from another round of revision. However, I won't object to accepting it if my co-reviewers champion it.

**Paper Topic And Main Contributions:**

The paper reports a CRF model aimed at predicting interlinear morphological glosses for both lexical and grammatical features.

**Questions For The Authors:**

The drop in performance for BERT needs to be explained, given that it is so much worse than CRF.

**Reasons To Accept:**

  1. a well designed experiment for the task
  2. an extension of a known CRF mechanism and its comparison to this baseline and to a BERT baseline.


**Reasons To Reject:**

  1. The task is fairly niche, which should not preclude its publication.
  2. I am not convinced about the multilingual aspect. From the title I expected a zero-shot transfer mechanism, for example, training on French, testing on Italian or German.


**Reproducibility:**

5: Could easily reproduce the results.

**Reviewer Confidence:**

4: Quite sure. I tried to check the important points carefully. It's unlikely, though conceivable, that I missed something that should affect my ratings.

---

> ### Author Rebuttal · Authors · 2023-08-26
>
> Thank you for your comments.
>
> On multilinguality, our goal was to see to what extent learning cross-linguistic features was possible, i.e. delexicalised features such as morpheme length or CV skeleton. By learning such features on a larger (and available) corpus, very low-resource languages may benefit, as in the last part of 5.5.
>
> A peculiarity of the task of automatic glossing is that cross-corpus and cross-lingual learning is made difficult by the annotation variability despite a general framework such as the Leipzig Glossing Rules: while lexical glosses (in English) vary depending on the topic and domains in each corpus, we also found that many grammatical glosses are in fact language-specific
>
> The BERT baseline is not ours but is provided by the shared-task organisers. As it seems, the base model (RoBERTa) is too complex to be efficiently fined-tuned with the small amount of data for most test languages. See also the discussion in Section 3.1 of https://arxiv.org/pdf/2303.14234.pdf. We will make sure to better explain this drop in our revised version.

---

### Meta-Review · Area_Chair_nTr8 · 2023-09-11

**Recommendation:** 4

**Metareview:**

This paper explores the concept of automatic glossing. The authors leverage not only the morpheme sequence of the target sentence but also its translation to determine the appropriate glossing. The authors introduce a latent variable Conditional Random Field (CRF) model, incorporating traditional symbol-based features. The authors place particular emphasis on addressing low-resource scenarios. They assert that their CRF model outperforms neural models.

This paper is well-written and the experiments were consistent with the research objectives. While it would not reach a large audience, this paper is worthy of presentation at the main conference.

---

### Decision · Program_Chairs · 2023-10-07

**Decision:**

Accept-Findings

**Comment:**

This paper explores the concept of automatic glossing. The authors leverage not only the morpheme sequence of the target sentence but also its translation to determine the appropriate glossing. The authors introduce a latent variable Conditional Random Field (CRF) model, incorporating traditional symbol-based features. The authors place particular emphasis on addressing low-resource scenarios. They assert that their CRF model outperforms neural models.

This paper is well-written and the experiments were consistent with the research objectives. While it would not reach a large audience, this paper is worthy of presentation at the main conference.